# Liraglutide Reduces Liver Steatosis and Improves Metabolic Indices in Obese Patients Without Diabetes: A 3-Month Prospective Study

**DOI:** 10.3390/ijms26125883

**Published:** 2025-06-19

**Authors:** Aleksandra Bołdys, Łukasz Bułdak, Michał Nicze, Bogusław Okopień

**Affiliations:** Department of Internal Medicine and Clinical Pharmacology, Faculty of Medical Sciences in Katowice, Medical University of Silesia, Medyków 18, 40-752 Katowice, Poland

**Keywords:** MASLD, liver steatosis, non-invasive markers, GLP-1 analog, obesity

## Abstract

Metabolic Dysfunction-Associated Steatotic Liver Disease (MASLD) is a leading cause of liver cirrhosis, with its global prevalence rising due to obesity, insulin resistance, and type 2 diabetes mellitus. While bariatric surgery remains effective for weight loss, Glucagon-Like Peptide-1 analogs such as liraglutide are emerging as promising pharmacological treatments. This study aimed to evaluate the effects of a 3-month liraglutide treatment on liver steatosis, subclinical markers, and insulin resistance in non-diabetic, obese patients with MASLD. Twenty-eight obese adults (BMI ≥ 30 kg/m^2^) were treated with daily subcutaneous liraglutide injections for three months. Liver steatosis was assessed using FibroScan^®^ (CAP score) and non-invasive indices (Hepatic Steatosis Index—HSI, and NAFLD Liver Fat Score—NLFS). Insulin resistance was measured with conventional markers (HOMA-IR, QUICKI) and triglyceride-based indices (METS-IR, TyG). Liraglutide significantly reduced liver steatosis (CAP score: 305 to 268 dB/m, *p* < 0.05) and improved HSI, while NLFS remained unchanged. Despite significant weight loss, traditional insulin resistance markers remained unchanged, while METS-IR and TyG improved. Liraglutide therapy improved liver steatosis and triglyceride-based insulin resistance markers in non-diabetic obese patients with MASLD. These findings support the use of liraglutide, highlighting the value of personalized approaches and alternative insulin resistance assessments in MASLD management.

## 1. Introduction

Metabolic Dysfunction-Associated Steatotic Liver Disease (MASLD), which previously was termed Non-Alcoholic Fatty Liver Disease (NAFLD), is one of the major causes of liver cirrhosis [1]. Its incidence is a growing healthcare issue in developed and developing countries [2,3]. It is estimated that by 2030, the number of cases of liver function decompensation, the incidence of hepatocellular carcinoma, and mortality due to complications of liver cirrhosis in the course of Metabolic Dysfunction-Associated Steatohepatitis (MASH) will increase 2–3 times [4]. MASLD is an independent factor of increasing morbidity and mortality from both hepatic causes and increased development of extrahepatic cancers.

It seems that the number of cases of MASLD is directly connected with the pandemic of obesity and its metabolic complications (insulin resistance, type 2 diabetes mellitus, arterial hypertension, lipid disorders). Currently, one in eight people around the world are obese [2].

It has been shown that effective lifestyle intervention results in significant improvements in the course of MASLD [5], including a reduction in the extent of liver steatosis, as well as in liver function tests (AST, ALT—aspartate and alanine aminotransferase, respectively) and inflammatory markers (Interferon gamma [IFN-γ], Tumor Necrosis Factor alpha [TNF-α]). Treatment of obesity with surgical procedures (sleeve gastrectomies and gastric bypasses) leads to a significant and long-standing reduction in weight, which is accompanied by a significant improvement in liver steatosis [6,7]. Bariatric surgery is considered a highly effective intervention for obesity; however, it is typically recommended for patients with advanced stages of the disease or with obesity-related complications [8].

Nowadays, there are several novel pharmacological therapeutic options that successfully reduce body weight, even by up to 20% in half of the treated patients [9,10]. The greatest expectations are connected with incretin-based therapies [11], which include Glucagon-Like Peptide-1 (GLP-1) analogs (e.g., liraglutide, semaglutide). Previous in vitro studies have shown a significant reduction in lipid accumulation in the hepatoma HepG2 cell line subjected to a GLP-1 receptor agonist [12].

The presence of fatty liver and its fibrosis are the predictive factors for disease progression [13,14]. There are several tools that may be employed in routine clinical practice to evaluate the risk of MASLD advancement. It has been shown that lifestyle modifications and therapeutic interventions can affect the early phase of MASLD, which is amenable to early detection with ultrasound-based diagnostic modalities [15]. Diagnostic algorithms are currently based largely on the use of clinical assessment, liver parameters, and ultrasound examination (as found in Polish guidelines) [16,17]. These tools facilitate the identification of advanced stages of MASLD and encompass the evaluation of disease-related risk factors, abnormal transaminase activity, ultrasound-based detection of hepatic steatosis, exclusion of other causes of steatosis, exclusion of advanced liver fibrosis, risk stratification based on the Fibrosis-4 (FIB-4) index, and elastographic assessment as well.

Therefore, we conducted a study to determine the efficacy of a 3-month weight management program, including dietary counseling and the initiation of GLP-1 analog (liraglutide) therapy, in relation to the degree of liver steatosis and subclinical markers of the disease, as well as prognostic algorithms used in the course of MASLD patients’ management.

## 2. Results

### 2.1. Baseline Characteristics

A total of 28 subjects (19 women and 9 men), as mentioned above, were included in the study and their baseline characteristics and post-treatment values are shown in Table 1 and Table 2.

### 2.2. The Impact of Liraglutide Therapy on Liver Steatosis (Measured by FibroScan^®^)

Obese patients demonstrated substantial liver steatosis, as assessed by FibroScan^®^ (CAP score), with a median value of 305 dB/m, indicative of stage 3 steatosis in the majority of cases (Appendix A). Following liraglutide treatment, a statistically significant reduction in CAP score was observed (*p* < 0.05), with a new median value of 268 dB/m. Although this numerical decrease corresponded to a shift to stage 2 steatosis, the change in the liver steatosis stage did not reach statistical significance (Table 3).

### 2.3. Impact of Liraglutide Therapy on Hepatic Steatosis Assessment Algorithms

While FibroScan^®^ is an extremely valuable tool for evaluating hepatic lipid accumulation, its relatively high cost may limit widespread clinical applicability. The effects of liraglutide treatment on diagnostic algorithms have been summarized in Table 4.

A significant improvement in the HSI level was noted, resulting in a reduction exceeding 5%. However, no notable change was observed in the NLFS.

### 2.4. Exploratory Markers in Liver Steatosis

Several exploratory laboratory markers have also been evaluated in the study as potential indicators of liver steatosis. Apolipoprotein A1, which is a scaffold protein of high-density lipoprotein (HDL), is known for its cardioprotective properties. However, findings to date suggest that apoA1 reduction corresponds with the later stages of MASLD [18,19]. Surprisingly, in our study cohort, a 3-month course of liraglutide led to a reduction in apoA1 levels. This decline mirrored the downward trend observed in HDL concentrations, although the latter did not achieve statistical significance (55.3 ± 15.8 vs. 52.99 ± 13.7 mg/dL; *p* = 0.11).

Another serum marker assessed in the study was a2M, previously reported to be associated with MASLD [20]. However, the 3-month therapy with a GLP-1 receptor agonist failed to result in any significant changes in a2M levels. Similarly, no alterations were observed in the concentration of TIMP1 throughout the duration of the study. The findings related to exploratory markers have been summarized in Table 5.

### 2.5. Impact on Body Weight and Markers of Insulin Resistance

GLP-1 agonists are currently approved for the treatment of obesity [9]. In this study, we demonstrate that even in the early phase of therapy, a significant reduction in body mass can be achieved following liraglutide administration. Additionally, we aimed to assess the degree of insulin resistance throughout the course of treatment (Table 6).

Interestingly, conventional insulin resistance indices, such as the Homeostasis Model Assessment of Insulin Resistance (HOMA-IR), and insulin sensitivity indices, such as the Quantitative Insulin Sensitivity Check Index (QUICKI), remained unchanged after 3 months of GLP-1 receptor agonist therapy. This may be explained by the fact that both indices are heavily dependent on insulin levels, which did not change significantly during the study period [17.30 (9.62–26.65) vs. 18.25 (11.87–24.43); *p* = 0.56]. In contrast, triglyceride-based models, namely the Metabolic Score for Insulin Resistance (METS-IR) and the Triglyceride–Glucose (TyG) index, demonstrated a reduction in insulin resistance by the end of the study. These findings underscore the importance of a comprehensive, multi-parameter approach when evaluating treatment outcomes and highlight the potential limitations of relying on a single surrogate marker.

### 2.6. Correlations

In our study, there were statistically significant correlations between weight loss and changes in HSI, QUICKI, METS-IR, and HOMA-IR, all of which suggest that body weight reduction is associated with improvements in liver fat and insulin sensitivity or resistance markers. The correlations are presented in Table 7.

### 2.7. Safety and Adverse Effects

During the 3-month treatment period, no cases of acute pancreatitis or other forms of acute abdominal pain were reported. Additionally, no asymptomatic elevations in serum amylase or lipase activity were observed. Five patients experienced nausea during the initial phase of therapy; however, this side effect resolved upon titration to the target dose of liraglutide.

## 3. Discussion

Our findings demonstrate a high prevalence of hepatic steatosis among obese individuals. A relatively short-term pharmacological intervention with liraglutide led to a significant reduction in the CAP score, with a mean decrease of 37 dB/m (approximately 12%). These outcomes are comparable to those reported for semaglutide in patients with NAFLD and type 2 diabetes mellitus, where CAP values decreased from 319 ± 37 to 283 ± 46 dB/m (*p* < 0.001) [21]. In non-diabetic obese subjects, our results appear only marginally less favorable than those achieved through bariatric surgery (341.0 vs. 277.0 dB/m at month 3; *p* < 0.001) [22].

One likely confounding factor is the more pronounced weight loss typically associated with bariatric procedures, along with the greater degree of baseline obesity in surgical cohorts (BMI > 40 kg/m^2^ or >35 kg/m^2^ with comorbidities). Furthermore, bariatric surgery exerts both mechanical (both restriction and malabsorption) as well as hormonal (incretin-mediated) effects on MASLD pathophysiology [23]. Although surgery can be highly effective, it carries inherent perioperative risks and long-term nutritional complications. In contrast, GLP-1 receptor agonists, such as liraglutide, offer a safer, pharmacological approach to improving metabolic profiles in obese patients.

Importantly, while weight reduction remains a cornerstone of MASLD management, evidence suggests that GLP-1 analogs also exert direct effects on hepatocytes, contributing to the attenuation of steatosis independently of weight loss [12], which constitutes a significant advantage over bariatric surgery by offering additional benefits with fewer potential complications.

Modern ultrasound-based techniques, such as FibroScan^®^, provide valuable tools for assessing liver steatosis and fibrosis [24]. However, their relatively high cost limits widespread implementation. The current imaging gold standard—MRI proton density fat fraction (MRI-PDFF)—offers excellent precision but is even less accessible due to cost and logistical barriers [25,26]. Consequently, several clinical algorithms incorporating demographic, anthropometric, and laboratory parameters have been developed to estimate hepatic fat content [27,28]. In our study, we employed two such algorithms—HSI and NLFS—both validated in NAFLD populations. Interestingly, they demonstrated divergent responses to treatment. Liraglutide therapy was associated with a significant reduction in HSI, while NLFS remained unchanged. This discrepancy likely reflects the differing variables used in each score. HSI is primarily influenced by body mass index, which decreased significantly over the study period, whereas NLFS is more dependent on insulin levels and the presence of diabetes mellitus—parameters that remained relatively stable. Notably, fasting insulin concentrations were not significantly altered during liraglutide treatment in our cohort, potentially explaining the lack of NLFS change. These findings align with preclinical and clinical data, indicating that GLP-1 receptor agonists may stimulate insulin expression [29] and elevate postprandial insulin levels without markedly affecting fasting concentrations [30]. Furthermore, improvements in insulin resistance markers observed during GLP-1 therapy appear to be largely mediated by weight reduction [31].

Our analysis also explored the utility of biochemical markers in monitoring MASLD progression. Simple, single-analyte markers, such as α2-macroglobulin (a2M) and tissue inhibitors of metalloproteinases-1 (TIMP1), did not show meaningful changes. Although we observed a decrease in apolipoprotein A1 (apoA1) levels, this was not paralleled by a statistically significant reduction in HDL cholesterol. These findings are in line with a recent meta-analysis on GLP-1 receptor agonists, which reported no significant effect on HDL levels (MD = −0.12, 95% CI: −0.73 to 0.49; *p* = 0.69) [32]. The discordance between apoA1 and HDL levels warrants further investigation and may be better understood by evaluating additional lipid-related ratios, such as apoB/apoA1 [33] or TG/apoA1, as demonstrated in the study by Wang et al. [34].

Insulin resistance is a central feature of MASLD, yet its assessment remains challenging due to the influence of demographic factors, comorbidities, and the lack of standardized cut-off values [35,36]. Although HOMA-IR is widely used in clinical practice, it primarily reflects β-cell function and is highly insulin-dependent [37]. In our study, HOMA-IR and QUICKI did not significantly change following therapy, likely due to stable fasting insulin levels. In contrast, triglyceride-based indices such as the TyG and METS-IR showed favorable reductions, supporting their utility in non-diabetic patients undergoing metabolic therapy.

Emerging evidence suggests that METS-IR may offer superior prognostic value, particularly in predicting cardiovascular risk in individuals under 65 years of age [38]. Therefore, early pharmacologic intervention aimed at improving METS-IR could yield long-term cardiometabolic benefits. Taken together, our data reinforce the notion that no single algorithm can universally capture the complexity of insulin resistance. In populations with metabolic syndrome features but without diabetes, indices such as METS-IR and TyG may provide more reliable insights compared to HOMA-IR or QUICKI.

The safety profile of Glucagon-Like Peptide-1 receptor agonists (GLP-1 RAs), including liraglutide, is of significant clinical relevance as these agents are increasingly used not only for type 2 diabetes but also for obesity management. Although in our study no significant serious adverse events appeared, the existing literature and post-marketing surveillance data emphasize the importance of cautious and individualized use. Among the GLP-1 RAs, liraglutide has been extensively studied and is associated with a well-characterized adverse event profile [39]. The most frequently reported side effects include gastrointestinal symptoms such as nausea, diarrhea, vomiting, decreased appetite, dyspepsia, and constipation. Immunogenic-related events have been reported in fewer than 0.8% of patients. More severe, though less common, adverse reactions include risk of C-cell tumors, acute pancreatitis, cholelithiasis, hypoglycemia (especially while combined with insulin), elevated heart rate, renal impairment, hypersensitivity reactions, suicidal ideation, and behavior. In addition to these established effects, recent pharmacovigilance studies have expanded our understanding of GLP-1 RA safety profiles. Kim et al. [40] conducted a disproportionality analysis using data from the global pharmacovigilance database VigiBase to compare adverse drug reaction patterns of liraglutide, semaglutide, and tirzepatide when used as anti-obesity agents. Their findings revealed that tirzepatide had the lowest overall frequency and strength of pharmacovigilance signals. Conversely, semaglutide was significantly associated with several unexpected and potentially serious adverse events, including suicidal ideation and behavior (adjusted reporting odds ratio [aROR] 2.52; information component lower limit [IC025] 1.28), hair loss (aROR 1.42; IC025 0.63), and vision loss (aROR 1.80; IC025 1.13).

## 4. Materials and Methods

### 4.1. Subjects

The study population consisted of 28 (19 females and 9 males) obese (BMI ≥ 30 kg/m^2^) adults, otherwise healthy participants, who were eligible for the treatment with a subcutaneous, daily injection of liraglutide (in a dose-ascending manner from 0.6 mg to 3 mg per day). The mean age of patients was 49.1 ± 11.6 years. The youngest participant was 26 and the oldest was 78. Patients were recruited from the Outpatient Clinic of the Department of Internal Medicine and Clinical Pharmacology (Medical University of Silesia, Katowice, Poland). All participants provided written informed consent to participate in the study. During the course of the experiments, two blood samplings were scheduled—prior to the therapy and three months after the initiation of liraglutide injections. All patients completed the study period. The ethical committee of the Medical University of Silesia had approved the study protocol (BNW/NWN/0052/KB1/97/23).

### 4.2. Inclusion and Exclusion Criteria

Inclusion criteria for the study were as follows: (a) Age ≥ 18 years; (b) Written informed consent from the volunteer to participate in the study; (c) Obesity (BMI ≥ 30 kg/m^2^); and (d) Ultrasound evidence of liver steatosis features on the first visit. Exclusion criteria included the following conditions: (a) Age under 18 years; (b) Lack of cooperation with the volunteer; (c) Lack of informed consent from the volunteer; (d) Pregnancy or breastfeeding; (e) History of hypersensitivity to semaglutide or dulaglutide; (f) Ketoacidosis; (g) Addiction to alcohol or psychoactive substances, including nicotine; (h) Acute and chronic inflammatory processes; (i) Abnormal parameters in laboratory tests: [eGFR < 60 mL/min/1.73 m^2^, ALT and AST > 3× ULN, total bilirubin > 1.2 mg/dL, Hgb < 10 g/dL or >16 g/dL, RBC < 3.5 M/μL or >5.5 M/μL, WBC < 3.5 K/μL or >10 K/μL, PLT < 140 K/μL or >400 K/μL]; (j) Currently used GLP-1 analog therapy; and (k) History of cancer (up to 5 years prior to inclusion).

### 4.3. Anthropometric and Laboratory Assessments

Patients qualified for the study underwent physical examination and anthropometric measurements, including height, body weight, and waist-to-hip ratio (WHR). Subsequently, a venous blood sample was collected from each participant to evaluate: complete blood count (2 mL), biochemical parameters (5 mL), including sodium, potassium, uric acid, urea, creatinine (with GFR estimation), liver function tests (total bilirubin, AST, ALT, GGT, ALP), glucose, lipid profile (total cholesterol, LDL, HDL, TG), TSH, ferritin, iron, and TIBC. All these analyses were conducted using the following laboratory equipment: Sysmex XN1000 (Sysmex Corporation, Kobe, Japan) device for complete blood count and Cobas PRO (Roche Diagnostics, Basel, Switzerland) device for biochemical tests. Portions of the obtained biological specimens (plasma—4 mL, serum—5 mL) were stored in a −70 °C freezer for further analysis of known and potential biomarkers of liver steatosis. These biomarkers included apolipoprotein A1 (apoA1) and alpha-2-macroglobulin (a2M). Both ApoA1 and a2M levels were determined using ELISA assays (Cloud-Clone Corp., Huston, TX, USA) and were performed according to the standard ELISA protocol. All reagents, samples, and standards were prepared prior to use. In total, 100 µL of either standard or sample was added to each well and incubated for 1 h at 37 °C. Following incubation, the wells were aspirated and 100 µL of detection reagent A was added, followed by a 1 h incubation at 37 °C. The wells were then aspirated and washed three times. Subsequently, 100 µL of detection reagent B was added and the plate was incubated for 30 min at 37 °C. Then, the wells were aspirated and washed five times. Subsequently, 90 µL of substrate solution was added to each well, and the plate was incubated for 10–20 min at 37 °C. Finally, 50 µL of stop solution was added, and absorbance was measured immediately using the xMark™ Microplate Absorbance Spectrophotometer at a wavelength of 450 nm (Bio-Rad, Hercules, CA, USA).

### 4.4. Liver Steatosis Assessment

Imaging tests used to assess liver steatosis and fibrosis include FibroScan^®^ (Echosens, Lyon, France), a non-invasive elastography method for screening and determining the degree of steatosis, measured by the Controlled Attenuation Parameter (CAP) and expressed in decibels (dB). Initially, this method was used solely to evaluate the severity of liver fibrosis; however, with advancements in technology, currently available systems also enable the assessment of liver steatosis. In the study, the level of steatosis was assessed using a FibroScan^®^ 430 Mini+ (Echosens, Lyon, France). It was performed after a 5 min rest in a lying position on the back, with the right upper limb under the head and the right lower limb placed over the left one. The transducer was positioned along the right mid-axillary line, at the level of the xiphoid process, in the intercostal space. During the procedure, preliminary measurements were taken to determine the optimal location, after which 10 measurements were recorded and utilized to evaluate the parameters under investigation.

The NAFLD Liver Fat Score (NLFS) [41] was calculated based on the presence of metabolic syndrome, type 2 diabetes mellitus, fasting serum insulin, fasting serum AST, and the AST/ALT ratio. The formula is as follows: NLFS = −2.89 + 1.18 × metabolic syndrome (yes = 1/no = 0) + 0.45 × type 2 diabetes mellitus (yes = 2/no = 0) + 0.15 × fasting serum insulin (mU/L) + 0.04 × fasting serum AST (UI/L)−0.94 × AST/ALT ratio.

The Hepatic Steatosis Index (HSI) was calculated using the following formula [42]: HSI = 8 × (ALT [UI/L]/AST [UI/L]) + BMI [kg/m^2^]. An additional 2 points are added if the patient is female or has diabetes mellitus.

### 4.5. Statistical Analysis

The minimum sample size was estimated to be 22 patients and calculated according to the assumption of the basal level of steatosis of 300 ± 25dB/m, with an approximate 10% reduction during the course of treatment. The probability of type I error was set at 0.05 with test power set at 80%. The collected data were analyzed with Statistica TIBCO Software Inc. (2017) version 13.3 software (Palo Alto, CA, USA). The normality of distribution in each dataset was evaluated using the Shapiro–Wilk’s test. Data that followed a normal distribution are presented as means with standard deviation, while data that did not follow a normal distribution are depicted as medians with interquartile ranges. Comparisons of data obtained before and after treatment were made using the paired samples *t*-test or the Wilcoxon signed-rank test. A *p*-value of <0.05 was considered statistically significant.

## 5. Conclusions

Short-term liraglutide therapy in obese patients without diabetes significantly reduces hepatic steatosis, as demonstrated by decreases in both the CAP score and Hepatic Steatosis Index (HSI). While these improvements are largely attributed to weight loss, emerging evidence suggests that GLP-1 receptor agonists may also exert direct, weight-independent hepatoprotective effects. In addition, improvements in insulin resistance were more accurately reflected by triglyceride-based indices (such as METS-IR and TyG) rather than traditional insulin-dependent markers (like HOMA-IR or QUICKI), underscoring the limitations of relying solely on fasting insulin levels. The proper selection of tools to assess liver steatosis is, therefore, essential in studies evaluating therapeutic strategies. Our findings support the use of GLP-1 receptor agonists as a viable, non-surgical option in the early treatment of MASLD. Notably, in obese individuals without diabetes, the most responsive diagnostic algorithms—METS-IR, TyG, and HSI—do not require insulin measurement, highlighting the need for a more individualized and multi-modal approach in monitoring treatment response and disease progression in this specific population.

However, it should be kept in mind that the use of GLP-1 receptor agonists requires clinical caution. While randomized controlled trials provide essential data on efficacy and safety, real-world evidence reveals rare or delayed adverse events that may not be apparent during pre-approval evaluation. This highlights the importance of ongoing pharmacovigilance, particularly as these agents are increasingly utilized in obesity management. Safe and effective use necessitates careful patient selection, individualized risk–benefit assessment, and proactive monitoring. Future research should aim to elucidate the mechanisms and risk factors underlying these adverse effects.

## Figures and Tables

**Table 1 ijms-26-05883-t001:** Changes in demographic characteristics between baseline and 3-month post-liraglutide therapy follow-up.

	Number of Patients	Age—Years (Mean ± SD)	BMI Before—kg/m^2^ (Mean ± SD or Median; Q1,Q3)	BMI After—kg/m^2^ (Mean ± SD or Median; Q1,Q3)	Body Weight Before—kg (Mean ± SD)	Body Weight After—kg (Mean ± SD or Median; Q1,Q3)	*p*-Value	WC Before—cm (Median; Q1,Q3)	WC After—cm (Median; Q1,Q3)	*p*-Value
Total	28	49.1 ± 11.6	35.63 ± 5.10	33.81 ± 4.97	102.32 ± 17.53	97.41 (89.28; 100)	<0.001	98.1 (94.4; 109.7)	96.2 (90.4; 103.5)	<0.001
Women	19	49.9 ± 12.9	35.27 ± 4.72	33.5 ± 4.65	96.68 ± 11.58	91.86 ± 11.78	<0.001	97.9 (93.7; 110.6)	96.1 (90.2; 103.0)	<0.001
Men	9	47.3 ± 8.6	35.06 (33.16; 36.39)	34.47 ± 5.83	114.22 ± 22.36	109.13 ± 21.98	<0.05	98.7 (95.5; 108.6)	96.3 (92.7; 104.7)	<0.05

The normality of data distribution was assessed using the Shapiro–Wilk’s test. Variables with normal distribution are presented as mean ± standard deviation (SD); non-normally distributed data are shown as median with interquartile range (IQR). Comparisons between values before and after the treatment were performed using the paired samples *t*-test (for normally distributed data) or the Wilcoxon signed-rank test (for non-normally distributed data). A *p*-value < 0.05 was considered statistically significant. Abbreviations and calculations: BMI—body mass index, calculated as the body mass divided by the square of the body height, and expressed in units of kg/m^2^; SD—standard deviation, Q1—first quartile, Q3—third quartile; WC—waist circumference.

**Table 2 ijms-26-05883-t002:** Baseline characteristics of laboratory tests in the study group before and after the liraglutide treatment.

**Parameter**	**Before Treatment**			**After Treatment**			***p*-Value**	**Reference Range/Desired Value**
	**Mean**	**SD**		**Mean**	**SD**			
TCh (mg/dL)	189.45	44.25		174.66	50.86		0.22	<190
LDL (mg/dL)	104.27	43.94		96.91	46.95		0.55	<115
HDL (mg/dL)	55.34	15.84		52.99	13.68		0.11	>60
TG (mg/dL)	151.09	86.91		121.44	58.47		<0.05	<150
nHDL (mg/dL)	134.10	44.93		121.67	48.82		0.32	<145
UA (mg/dL)	6.36	1.46		6.08	1.47		0.09	2.40–5.70
Cr (mg/dL)	0.87	0.13		0.89	0.15		0.21	0.51–0.95
Na (mmol/L)	139.14	1.54		140.11	2.11		<0.05	136.0–145.0
RBC (10^6^/μL)	4.90	0.53		4.85	0.58		0.74	3.7–5.0
Hgb (g/dL)	14.57	1.70		14.78	1.53		0.16	11.5–15.0
PLT (10^3^/μL)	267.89	62.25		289.39	67.83		0.11	130–400
	**Median**	**Q1**	**Q3**	**Median**	**Q1**	**Q3**	** *p* ** **-Value**	
ALT (UI/mL)	30.20	22.15	47.80	28.15	21.53	43.33	0.07	<35.0
AST (UI/mL)	23.75	20.78	33.45	24.55	20.35	33.48	0.52	<35.0
GGT (UI/mL)	27.90	20.93	39.73	22.85	16.19	35.13	<0.05	<40
Bil (mg/dL)	0.48	0.39	0.72	0.46	0.37	0.59	<0.05	0.30–1.20
ALP (UI/mL)	63.00	60.00	79.25	65.00	55.50	75.25	0.06	35–104
HbA1c (%)	5.65	5.33	5.93	5.53	5.23	5.95	<0.05	4.80–5.90
Insulin (µU/mL)	17.30	9.62	26.65	18.25	11.87	24.43	0.56	2.6–24.9
Glu (mg/dL)	98.45	88.90	105.00	90.85	83.48	100	<0.05	70.00–99.00
K (mmol/L)	4.33	4.10	4.55	4.36	4.20	4.50	0.71	3.50–5.10
Cl (mmol/L)	102.95	101.15	102.70	102.20	100.60	103.40	0.32	98.00–107.00
MCV (fL)	88.90	87.30	91.60	90.75	87.53	92.70	<0.05	84–98

The normality of data distribution was assessed using the Shapiro–Wilk’s test. Variables with normal distribution are presented as mean ± standard deviation (SD); non-normally distributed data are shown as median with interquartile range (IQR). Comparisons between values before and after the treatment were performed using the paired samples *t*-test (for normally distributed data) or the Wilcoxon signed-rank test (for non-normally distributed data). A *p*-value < 0.05 was considered statistically significant. Abbreviations: TCh—Total Cholesterol, LDL—Low-Density Lipoprotein Cholesterol, HDL—High-Density Lipoprotein Cholesterol, TG—Triglycerides, nHDL—non-HDL Cholesterol, Cr—Creatinine, UA—Uric Acid, Na—Natrium, RBC—Red Blood Cells, Hgb—Hemoglobin, PLT—Platelets, ALT—Alanine Aminotransaminase, AST—Aspartate Aminotransferase, GGT—Gamma-glutamyl Transferase, Bil—Bilirubine, ALP—Alkaline Phosphatase, HbA1c—Glycated Hemoglobin, Glu—Glucose, K—Potassium, Cl—Chloride, MCV—Mean Corpuscular Volume.

**Table 3 ijms-26-05883-t003:** Effect of Glucagon-Like Peptide-1 (GLP-1) agonist on liver steatosis measured by FibroScan^®^.

Parameter	Before Treatment			After Treatment			*p*-Value
	**Median**	**Q1**	**Q3**	**Median**	**Q1**	**Q3**	
CAP (dB/m)	305	232.5	325.5	268	241.5	297.0	<0.05
steatosis	3	1	3	2	1	3	0.08

The normality of data distribution was assessed using the Shapiro–Wilk’s test. Non-normally distributed data are shown as median with interquartile range (IQR). Comparisons between values before and after treatment were performed using the Wilcoxon signed-rank test. A *p*-value < 0.05 was considered statistically significant. Abbreviation: CAP—Controlled Attenuation Parameter^®^.

**Table 4 ijms-26-05883-t004:** Effect of Glucagon-Like Peptide-1 (GLP-1) agonists on steatotic parameters and predictive models.

Parameter	Before Treatment			After Treatment			*p*-Value
	**Mean**	**SD**		**Mean**	**SD**		
HSI	47.38	5.92		44.99	5.85		<0.001
	**Median**	**Q1**	**Q3**	**Median**	**Q1**	**Q3**	
NLFS	1.83	0.12	3.04	1.35	0.08	2.84	0.91

The normality of data distribution was assessed using the Shapiro–Wilk’s test. Variables with normal distribution are presented as mean ± standard deviation (SD); non-normally distributed data are shown as median with interquartile range (IQR). Comparisons between values before and after the treatment were performed using the paired samples *t*-test (for normally distributed data) or the Wilcoxon signed-rank test (for non-normally distributed data). A *p*-value < 0.05 was considered statistically significant. Abbreviations and calculations: HSI—Hepatic Steatosis Index, calculated as: 8 × (ALT/AST) + BMI + 2 (if diabetic) + 2 (if female), NLFS—Non-Alcoholic Fatty Liver Disease (NAFLD) Liver Fat Score, calculated as: 2.89 + 1.18 × metabolic syndrome (yes = 1/no = 0) + 0.45 × type 2 diabetes mellitus (yes = 2/no = 0) + 0.15 × fasting serum insulin (mUI/L) + 0.04 × fasting serum AST (UI/L) − 0.94 × AST/ALT ratio.

**Table 5 ijms-26-05883-t005:** Effect of Glucagon-Like Peptide-1 (GLP-1) agonist on exploratory markers in liver steatosis.

Parameter	Before Treatment			After Treatment			*p*-Value
	**Median**	**Q1**	**Q3**	**Median**	**Q1**	**Q3**	
APOA1 (µg/mL)	73.60	60.60	83.20	67.30	55.90	80.25	<0.05
TIMP1 (µg/mL)	129.35	99.50	150.90	131.35	94.25	145.15	0.16
a2M (µg/L)	223.30	180.85	283.88	246.80	204.15	353.63	0.2

The normality of data distribution was assessed using the Shapiro–Wilk’s test. Non-normally distributed data are shown as median with interquartile range (IQR). Comparisons between values before and after the treatment were performed using the Wilcoxon signed-rank test. A *p*-value < 0.05 was considered statistically significant. Abbreviations: APOA1—Apolipoprotein A1, TIMP1—Tissue Inhibitors of Metalloproteinase 1, a2M—Alpha-2-Macroglobulin.

**Table 6 ijms-26-05883-t006:** Effect of Glucagon-Like Peptide-1 (GLP-1) agonist on selected markers of insulin resistance before and after the treatment.

Parameter	Before Treatment			After Treatment			*p*-Value
	**Mean**	**SD**		**Mean**	**SD**		
QUICKI	0.31	0.04		0.31	0.04		0.79
METS-IR	53.01	10.93		49.68	10.13		<0.001
TyG	8.82	0.68		8.56	0.47		<0.05
	**Median**	**Q1**	**Q3**	**Median**	**Q1**	**Q3**	
HOMA-IR	4.36	2.38	6.92	4.09	2.49	6.24	0.69

The normality of data distribution was assessed using the Shapiro–Wilk’s test. Variables with normal distribution are presented as mean ± standard deviation (SD); non-normally distributed data are shown as median with interquartile range (IQR). Comparisons between values before and after the treatment were performed using the paired samples *t*-test (for normally distributed data) or the Wilcoxon signed-rank test (for non-normally distributed data). A *p*-value < 0.05 was considered statistically significant. Abbreviations and calculations: SD—standard deviation; Q1—first quartile; Q3—third quartile; QUICKI—Quantitative Insulin Sensitivity Check Index, calculated as: 1/[log(fasting insulin) + log(fasting glucose)]; METS-IR—Metabolic Score for Insulin Resistance, calculated as: (ln[2 × fasting glucose (mg/dL) + triglycerides (mg/dL)] × BMI (kg/m^2^))/ln[HDL-C (mg/dL)]; TyG—Triglyceride-Glucose index, calculated as: ln(fasting triglycerides (mg/dL) × fasting glucose (mg/dL)/2); HOMA-IR—Homeostasis Model Assessment of Insulin Resistance, calculated as: (fasting glucose (mg/dL) × fasting insulin (μU/mL)/405.

**Table 7 ijms-26-05883-t007:** Correlations for predictive models of steatosis.

	%Δ Body Weight	
	R	*p*
%Δ HSI	0.413	<0.05
%Δ QUICKI	−0.424	<0.05
%Δ METS-IR	0.694	<0.001
%Δ TyG	0.282	0.147
%Δ HOMA-IR	0.426	<0.05
%Δ NLFS	0.034	0.418

Correlations were calculated using Spearman’s or Pearson’s correlation coefficients, depending on the distribution of the data. The analysis was performed using the delta values (%Δ), representing the percentage change between baseline and post-treatment measurements. Statistical significance was set at *p* < 0.05. Abbreviations and calculations: %Δ—percentage of delta (change), R—correlation coefficient; *p*—significance level; HSI—Hepatic Steatosis Index, calculated as: 8 × (ALT/AST) + BMI + 2 (if diabetic) + 2 (if female); QUICKI—Quantitative Insulin Sensitivity Check Index, calculated as: 1/[log (fasting insulin) + log (fasting glucose)]; METS-IR—Metabolic Score for Insulin Resistance, calculated as: (ln[2 × fasting glucose (mg/dL) + triglycerides (mg/dL)] × BMI (kg/m^2^))/ln[HDL-C (mg/dL)]; TyG—Triglyceride-Glucose index, calculated as: ln(fasting triglycerides (mg/dL) × fasting glucose (mg/dL)/2); HOMA-IR—Homeostasis Model Assessment of Insulin Resistance, calculated as: (fasting glucose (mg/dL) × fasting insulin (μU/mL))/405; NLFS—Non-Alcoholic Fatty Liver Disease (NAFLD) Liver Fat Score, calculated as: 2.89 + 1.18 × metabolic syndrome (yes = 1/no = 0) + 0.45 × type 2 diabetes mellitus (yes = 2/no = 0) + 0.15 × fasting serum insulin (mU/L) + 0.04 × fasting serum AST (UI/L) − 0.94 × AST/ALT ratio.

## Data Availability

Interested researchers may request data from the corresponding authors, pending a reasonable justification.

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
