# Peer review of "Liraglutide Reduces Liver Steatosis and Improves Metabolic Indices in Obese Patients Without Diabetes: A 3-Month Prospective Study"

_ijms, 2025, doi:10.3390/ijms26125883_

Round 1
Reviewer 1 Report
Comments and Suggestions for Authors
In the manuscript ijms-3703693, the authors have determined the beneficial effects of liraglutide on liver steatosis and triglyceride-based indices in obesity without diabetes. This manuscript has strength in conducting a clinical trial and demonstrating the metabolic effects of liraglutide through multiple analyses. However, there are some issues required to be addressed.
- In Table 1, please show waist circumference (WC) data of the participants. Since abdominal obesity exerts greater effects on metabolic dysregulation, WC is also critical as well as BMI.
- Please show the BMI, body weight (Table 1), and blood parameters (Table 2) both before and after the liraglutide treatment.
- I wonder if the authors can show visualized data of liver steatosis. It will be helpful to capture the liraglutide effect on liver steatosis.
- Please make the tables stand alone. For example, in the legend of Table 6, please show how the parameters are calculated. Also, for all the tables, please mention the method of statistical analyses.
- Although liraglutide exerted beneficial effects on metabolism in the current manuscript, side effects of GLP-1 analogues have recently been reported. Please discuss about these at the end of the manuscript to prevent misuse or overuse of GLP-1 analogues.
Author Response
PLEASE SEE THE ATTACHMENT
We would like to express our sincere gratitude to the Reviewer for their insightful and thorough review of our manuscript. The comments and suggestions provided were highly valuable and have significantly contributed to improving the clarity and overall quality of the paper. We greatly appreciate the time and effort invested in the review process.
Please find attached our detailed responses to the Reviewer’s comments in the rebuttal letter. All changes to the manuscript are marked in red, and our responses are highlighted in green.
The revised manuscript has also been submitted.

Reviewer 2 Report
Comments and Suggestions for Authors
Authors prospectively evaluated the therapeutic effect of liraglutide for liver steatosis. As authors mentioned, the number or significance of MASLD was increasing. Furthermore, effective pharmacological interventions for MASLD is limited. Therefore, this study is interesting and valuable. Several issues remained to be addressed.
- In this study, most of patients showed normal ALT. Were they MASLD?
- In this study, the changes of CAP were shown. The changes of liver stiffness should be also shown.
- The changes of liver parameters such as AST, ALT or GGT should be also shown.
- Body weight change should be also shown.
Author Response
PLEASE SEE THE ATTACHMENT
We would like to sincerely thank the Reviewer for their thorough and insightful review, as well as
for the valuable comments and questions raised, particularly regarding the clarification of the inclusion
criteria. Please find our detailed responses below (changes are marked in red).
We hope they will adequately address the Reviewer’s concerns and provide clarification on the matters addressed by the Reviewer.
Please find below our detailed responses to the Reviewer’s comments included in the rebuttal letter, which are highlighted in green. All corresponding changes made in the manuscript are marked in red.
The revised manuscript has also been submitted.

Round 2
Reviewer 2 Report
Comments and Suggestions for Authors
Revised manuscript was well-addressed to the reviewer's comment or suggestion and well-written. It is an informative work to address the therapeutic effects of liraglutide for liver steatosis.